# DataInf: Efficiently Estimating Data Influence in LoRA-tuned LLMs and Diffusion Models

**Yongchan Kwon**[1,†], **Eric Wu**[1,§], **Kevin Wu**[1,§] **& James Zou**[§*]
Columbia University[†], Stanford University[§]

## Abstract

Quantifying the impact of training data points is crucial for understanding the outputs of machine learning models and for improving the transparency of the AI pipeline. The influence function is a principled and popular data attribution method, but its computational cost often makes it challenging to use. This issue becomes more pronounced in the setting of large language models and text-to-image models. In this work, we propose DataInf, an efficient influence approximation method that is practical for large-scale generative AI models. Leveraging an easy-to-compute closed-form expression, DataInf outperforms existing influence computation algorithms in terms of computational and memory efficiency. Our theoretical analysis shows that DataInf is particularly well-suited for parameter-efficient fine-tuning techniques such as LoRA. Through systematic empirical evaluations, we show that DataInf accurately approximates influence scores and is orders of magnitude faster than existing methods. In applications to RoBERTa-large, Llama-2-13B-chat, and stable-diffusion-v1.5 models, DataInf effectively identifies the most influential fine-tuning examples better than other approximate influence scores. Moreover, it can help to identify which data points are mislabeled.

## 1 Introduction

Modern large language models (LLMs) and text-to-image models have demonstrated remarkable abilities in generating human-like texts and photorealistic images, leading to diverse real-world applications such as translation, dialogue systems, and image editing (Brown et al., 2020; Rombach et al., 2022; Jiao et al., 2023). Nevertheless, even state-of-the-art models generate factually incorrect predictions or even biased outputs (Abid et al., 2021; Ouyang et al., 2022; Ferrara, 2023), often as a result of issues in the training data. This highlights the need for principled and systematic methods to quantify the impact of specific training data points. The influence function provides a rigorous framework for evaluating the impact of each training data point on model predictions (Hampel, 1974; Cook & Weisberg, 1980). Its efficacy has been demonstrated across various downstream machine learning tasks: mislabeled data detection (Koh & Liang, 2017), best subset selection (Feldman & Zhang, 2020; Guo et al., 2021), model interpretation (Han et al., 2020; Aamir et al., 2023; Grosse et al., 2023), and investigation of model biases (Wang et al., 2019; Kong et al., 2022).

While the influence function has shown promising results, its application in real-world scenarios poses practical challenges because of its expensive computational costs. Calculating the influence function requires the computation of the inverse Hessian matrix, which involves intensive computation. Previous studies have attempted to reduce this burden; however, most existing methods still require an iterative algorithm (Martens, 2010; Agarwal et al., 2017), multiple eigenvalue decompositions (George et al., 2018) or the training of numerous models (Feldman & Zhang, 2020) to obtain accurate influence estimates. It has therefore been very challenging to compute the influence function for large models such as LLMs (Devlin et al., 2018; Liu et al., 2019; Touvron et al., 2023) and diffusion models (Sohl-Dickstein et al., 2015; Ho et al., 2020; Rombach et al., 2022).

**Our contributions** We propose DataInf, a computationally efficient influence approximation method that can be easily applied to large-scale machine learning models. DataInf is based on an easy-to-compute closed-form expression, leading to better computational and memory complexities

---

*The first three authors equally contributed. Corresponding author: James Zou, jamesz@stanford.edu.

than existing state-of-the-art influence computation algorithms. Our approximation error analysis suggests that DataInf is especially effective when it is applied to parameter-efficient fine-tuned models. We evaluate the practical efficacy of DataInf through three sets of experiments: approximation error analysis, mislabeled data detection, and influential data identification. Our empirical results demonstrate that DataInf is faster and more effective in retrieving the most (or the least) influential training data points than existing algorithms. We apply DataInf to the RoBERTa, Llama-2-13B-chat, and stable-diffusion-v1.5 models, demonstrating that it is easily applicable to LLMs and large-scale diffusion models. Python-based implementation codes are available at `https://github.com/ykwon0407/DataInf`.

## 2 PRELIMINARIES

We denote an input space and a label space by $\mathcal{X}$ and $\mathcal{Y}$, respectively. We denote a training dataset by $\mathcal{D} = \{(x_i, y_i)\}_{i=1}^{n}$ where $x_i \in \mathcal{X}$ and $y_i \in \mathcal{Y}$ are an input and a label of the $i$-th datum. We consider the empirical risk minimization framework: For a loss function $\ell : \mathcal{Y} \times \mathcal{Y} \to \mathbb{R}$ and a parameter space $\Theta$, the empirical risk minimizer is defined as follows: $\theta^* := \mathrm{argmin}_{\theta \in \Theta} \frac{1}{n} \sum_{i=1}^{n} \ell(y_i, f_\theta(x_i))$ where $f_\theta : \mathcal{X} \to \mathcal{Y}$ is a model parametrized with $\theta \in \Theta$. We set $[m] := \{1, \dots, m\}$ for $m \in \mathbb{N}$. For $i \in [n]$ and a vector $\eta$, we denote a gradient of the $i$-th data point's loss with respect to $\eta$ by $\nabla_\eta \ell_i := \nabla_\eta \ell(y_i, f_\theta(x_i))$.

### 2.1 INFLUENCE FUNCTION

The influence function assesses the impact of individual training data points on parameter estimation (Hampel, 1974; Cook & Weisberg, 1980; Martin & Yohai, 1986). It captures how fast parameter estimates change when a particular data point is up-weighted. To be more specific, for $k \in [n]$ and $\varepsilon \in \mathbb{R}$, we consider the following $\varepsilon$-weighted risk minimization problem: $\theta^{(k)}(\varepsilon) := \mathrm{argmin}_{\theta \in \Theta} \frac{1}{n} \sum_{i=1}^{n} \ell(y_i, f_\theta(x_i)) + \varepsilon \ell(y_k, f_\theta(x_k))$. When a loss function $\ell(y, f_\theta(x))$ is twice-differentiable and strongly convex in $\theta$ for all $(x, y) \in \mathcal{X} \times \mathcal{Y}$, the empirical risk minimizer $\theta^*$ is well-defined, and the influence of the $k$-th data point $(x_k, y_k) \in \mathcal{D}$ on the empirical risk minimizer $\theta^*$ is defined as the derivative of $\theta^{(k)}(\varepsilon)$ at $\varepsilon = 0$:

$$\mathcal{I}_{\theta^*}(x_k, y_k) := \left. \frac{d\theta^{(k)}}{d\varepsilon} \right|_{\varepsilon=0} = -H(\theta^*)^{-1} \nabla_\theta \ell_k,$$

where $H(\theta) := \nabla_\theta^2 \left( n^{-1} \sum_{i=1}^{n} \ell(y_i, f_\theta(x_i)) \right)$ is the Hessian of the empirical loss (Hampel, 1974; Van der Vaart, 2000).

In machine learning problems, the influence function $\mathcal{I}_{\theta^*}(x_k, y_k)$ on the empirical risk minimizer $\theta^*$ is extended to the influence function on a prediction loss (Koh & Liang, 2017). For a validation dataset $\mathcal{D}^{(\mathrm{val})} := \{(x_i^{(\mathrm{val})}, y_i^{(\mathrm{val})})\}_{i=1}^{m}$, the influence of $(x_k, y_k)$ on the validation loss is defined as:

$$\mathcal{I}(x_k, y_k) := \left( \frac{1}{m} \sum_{i=1}^{m} \nabla_\theta \ell(y_i^{(\mathrm{val})}, f_\theta(x_i^{(\mathrm{val})}))|_{\theta=\theta^*} \right)^T \mathcal{I}_{\theta^*}(x_k, y_k).$$

The influence function $\mathcal{I}(x_k, y_k)$ provides an intuitive interpretation of how one data point affects the validation loss. When $\mathcal{I}(x_k, y_k)$ is a large positive (*resp.* negative) value, the validation loss would increase (*resp.* decrease) as the data point $(x_k, y_k)$ is up-weighted because $\mathcal{I}(x_k, y_k)$ is defined as a gradient of the validation loss. In other words, the influence function intuitively represents whether $(x_k, y_k)$ is beneficial or detrimental to the prediction loss.

While the influence function is established on a rigorous statistical framework, its computation often poses practical challenges due to the second-order gradients in $H(\theta^*)$. Calculating the second-order gradient is computationally intensive in general, but it can be achieved with the first-order gradient when the loss function is a negative log-likelihood function (Bartlett, 1953). To elaborate, suppose $\ell(y, f_\theta(x)) = -\log p(y \mid f_\theta(x))$ for all $(x, y) \in \mathcal{X} \times \mathcal{Y}$ and $\theta \in \Theta$ where $p(y \mid f_\theta(x))$ is a probability density function of $(x, y)$ at $\theta$. Bartlett's second identity implies that

$$\mathbb{E}\left[ \nabla_\theta^2 \ell(Y, f_\theta(X)) \right] = \mathbb{E}\left[ \nabla_\theta \ell(Y, f_\theta(X)) \left( \nabla_\theta \ell(Y, f_\theta(X)) \right)^T \right],$$

where the expectation is over the distribution $p(Y \mid f_\theta(X))$. That is, the Hessian $H(\theta^*)$ can be replaced with the second moment of the first-order gradients $G(\theta^*) := n^{-1} \sum_{i=1}^n \nabla_\theta \ell_i \nabla_\theta \ell_i^T$. This yields the following formula for the influence function:

$$-\left( \frac{1}{m} \sum_{i=1}^m \nabla_\theta \ell(y_i^{(\text{val})}, f_\theta(x_i^{(\text{val})}))|_{\theta=\theta^*} \right)^T G(\theta^*)^{-1} \nabla_\theta \ell_k. \tag{1}$$

In this paper, we restrict our focus to a negative log-likelihood function to leverage a simplified form of the Hessian function. A negative log-likelihood function is one of the most commonly used loss functions and many LLMs are pre-trained with the cross-entropy loss function, which is equivalent to a negative log-likelihood in classification problems (Touvron et al., 2023).

## 2.2 INFLUENCE FUNCTION FOR DEEP NEURAL NETWORK MODELS

The influence function in equation 1 can be computed with only the first-order gradients; however, there are practical challenges when $f_\theta$ is a deep neural network model (Basu et al., 2020; Bae et al., 2022). First, when the dimension of $\theta$ exceeds the sample size $n$, which is common in many modern machine learning problems, $G(\theta)$ is not invertible because the rank of $G(\theta)$ is at most $n$. Second, the size of $G(\theta)$ is too large to compute, making its computation infeasible.

To address the first issue, the "damping Hessian" approach is used in which a small positive number is added to diagonal elements of $G(\theta)$ and make it positive definite (Martens, 2010). As for the second issue, $G(\theta)$ is replaced with its block diagonal matrix, where each block corresponds to a layer of a deep neural network model. To be more specific, suppose $f_\theta$ can be expressed as a composition function $f_\theta(x) = f_{\theta_L} \circ \cdots \circ f_{\theta_1}(x)$ where for $l \in [L]$, we denote a vectorized notation of weights and biases in the $l$-th layer by $\theta_l \in \mathbb{R}^{d_l}$ for some $d_l \in \mathbb{N}$. Then, the $l$-th diagonal block of $G(\theta)$ can be expressed as $G_l(\theta) := n^{-1} \sum_{i=1}^n \nabla_{\theta_l} \ell_i \nabla_{\theta_l} \ell_i^T$, and $G(\theta)$ is replaced with $\text{diag}(G_1(\theta), \ldots, G_L(\theta))$ (Grosse et al., 2023). Combining these approaches gives the following influence function:

$$-\sum_{l=1}^L v_l^T \left( G_l(\theta^*) + \lambda_l I_{d_l} \right)^{-1} \nabla_{\theta_l} \ell_k \tag{2}$$

where $v_l := m^{-1} \sum_{i=1}^m \nabla_{\theta_l} \ell(y_i^{(\text{val})}, f_\theta(x_i^{(\text{val})}))|_{\theta=\theta^*}$, $\lambda_l$ is some positive constant, and $I_{d_l} \in \mathbb{R}^{d_l \times d_l}$ is the identity matrix of size $d_l$. The influence function in equation 2 not only stabilizes but also simplifies the computation of the Hessian matrix, becoming the standard estimand in the literature.

Shifting the focus of the influence function from equation 1 to equation 2 makes the calculation more feasible, yet it is often costly, especially when $d_l$ is large. We next review one of the most widely used approximate methods called LiSSA.

**LiSSA** Agarwal et al. (2017) proposed an iterative approach to compute the inverse Hessian vector product $\left( G_l(\theta^*) + \lambda_l I_{d_l} \right)^{-1} v_l$. For $r_{l,0} = v_l$, LiSSA recursively computes the following equation: $r_{l,j} = v_l + (I - (G_l(\theta^*) + \lambda_l I_{d_l})) r_{l,j-1}$. Agarwal et al. (2017) showed that when $(G_l(\theta^*) + \lambda_l I_{d_l}) \preceq I_{d_l}$ in the Löwner order, the $r_{l,j}$ converges to $(G_l(\theta^*) + \lambda_l I_{d_l})^{-1} v_l$ as $j$ increases. The influence function based on LiSSA is obtained by computing $-\sum_{l=1}^L r_{l,j}^T \nabla_{\theta_l} \ell_k$. In essence, it uses the following approximation:

$$r_{l,j} \approx (G_l(\theta^*) + \lambda_l I_{d_l})^{-1} v_l. \tag{3}$$

In practice, it is often assumed that LiSSA converges to the inverse Hessian vector product $\left( G_l(\theta^*) + \lambda_l I_{d_l} \right)^{-1} v_l$ in a reasonable number of iterations. When the number of iterations is finite, the computational complexity for LiSSA becomes $O(\sum_{l=1}^L n d_l^2)$ operations with $O(\max_{l \in [L]} d_l^2)$ memory complexity.[1]

---

[1]The computational complexity can be further accelerated to $O(\sum_{l=1}^L n d_l)$ with improved memory complexity $O(\max_{l \in [L]} d_l)$ by leveraging the first-order gradients. These big $O$ complexities are equal to those of the proposed method, but ours still has advantages over LiSSA as it does not require an expensive iterative algorithm. In our experiments, we compare ours with this accelerated LiSSA algorithm.

Several approaches, including LiSSA, have been studied to efficiently compute the influence function for deep neural network models. However, most of the existing methods require expensive iterative algorithms (Koh & Liang, 2017; Schioppa et al., 2022), multiple eigenvalue decomposition operations (Grosse et al., 2023), or the training of a numerous number of models (Feldman & Zhang, 2020). Consequently, when attempting to apply these methods to LLMs or diffusion models, their feasibility becomes severely constrained. In response to this significant challenge, we introduce a new closed-form expression that approximates the influence function.

## 3 DATAINF: EFFICIENT INFLUENCE COMPUTATION

We propose DataInf, an efficient influence computation algorithm characterized by an easy-to-compute closed-form expression. DataInf has better efficiency in both computational and memory complexities than existing state-of-the-art methods. The key approximation of DataInf is to swap the order of the matrix inversion and the average calculations in $\left(G_l(\theta^*) + \lambda_l I_{d_l}\right)^{-1}$ as follows:

$$\left(\frac{1}{n}\sum_{i=1}^{n}\nabla_{\theta_l}\ell_i\nabla_{\theta_l}\ell_i^T + \lambda_l I_{d_l}\right)^{-1} \approx \frac{1}{n}\sum_{i=1}^{n}\left(\nabla_{\theta_l}\ell_i\nabla_{\theta_l}\ell_i^T + \lambda_l I_{d_l}\right)^{-1}. \quad (4)$$

Here, the term $\left(\nabla_{\theta_l}\ell_i\nabla_{\theta_l}\ell_i^T + \lambda_l I_{d_l}\right)^{-1}$ has a closed-form expression because it is an inverse of the sum of a rank-one matrix and a diagonal matrix. To be more specific, leveraging the Sherman-Morrison formula, the right-hand side of equation 4 can be simplified as follows:

$$\frac{1}{n}\sum_{i=1}^{n}\left(\nabla_{\theta_l}\ell_i\nabla_{\theta_l}\ell_i^T + \lambda_l I_{d_l}\right)^{-1} = \frac{1}{n\lambda_l}\sum_{i=1}^{n}\left(I_{d_l} - \frac{\nabla_{\theta_l}\ell_i\nabla_{\theta_l}\ell_i^T}{\lambda_l + \nabla_{\theta_l}\ell_i^T\nabla_{\theta_l}\ell_i}\right).$$

In short, the inverse Hessian part, the left-hand side of equation 4, can be approximated with a closed-form expression. Based on this finding, we propose DataInf that efficiently approximates the influence function as follows.

$$\mathcal{I}_{\text{DataInf}}(x_k, y_k) = \sum_{l=1}^{L}\frac{1}{\lambda_l}\left(\frac{1}{n}\sum_{i=1}^{n}\frac{L_{l,i}}{\lambda_l + L_{l,ii}}L_{l,ik} - L_{l,k}\right), \quad (5)$$

where $L_{l,ij} := \nabla_{\theta_l}\ell_i^T\nabla_{\theta_l}\ell_j \in \mathbb{R}$ for all $l \in [L]$ and $i, j \in [n]$ and $L_{l,i} := v_l^T\nabla_{\theta_l}\ell_i \in \mathbb{R}$ for all $l \in [L]$ and $i \in [n]$. The equation 5 provides easy-to-compute expression of $\mathcal{I}_{\text{DataInf}}(x_k, y_k)$. We provide a pseudo algorithm in Appendix A.

DataInf can be computed in $O(\sum_{l=1}^{L}nd_l)$ operations with $O(\max_{l\in[L]}d_l)$ memory. In terms of computational complexity, DataInf is much faster than LiSSA, and it does not require iterative operations. Moreover, DataInf has a better memory complexity than LiSSA because it does not require storing Hessian matrices. Table 1 compares DataInf with the exact computation of the influence function (equation 2, denoted by Exact) and LiSSA when a model is a multilayer perceptron.

**Approximation error analysis** While the approximation in equation 4 provides an efficient computation method, it may exhibit significant errors because the two terms are not equal in general. To this end, we theoretically investigate approximation error incurred by equation 4. To elaborate, we set $S_{li} := \nabla_{\theta_l}\ell_i\nabla_{\theta_l}\ell_i^T + \lambda_l I_{d_l}$. The $l$-th part of the influence function in equation 2 can be expressed as $-v_l^T\left(n^{-1}\sum_{i=1}^{n}S_{li}\right)^{-1}\nabla_{\theta_l}\ell_k$, and that of the proposed method is $-v_l^T\left(n^{-1}\sum_{i=1}^{n}S_{li}^{-1}\right)\nabla_{\theta_l}\ell_k$. Then, the difference between these two terms is bounded by $\|v_l\|_2\left\|\left(\frac{1}{n}\sum_{i=1}^{n}S_{li}\right)^{-1} - \frac{1}{n}\sum_{i=1}^{n}S_{li}^{-1}\right\|_2\|\nabla_{\theta_l}\ell_k\|_2$. Here, we denote the spectral norm of a matrix $A$ by $\|A\|_2$ and denote the $L^2$ norm of a vector $v$ by $\|v\|_2$. In summary, the approximation error mainly depends on the spectral norm of the difference $\left(\left(n^{-1}\sum_{i=1}^{n}S_{li}\right)^{-1} - n^{-1}\sum_{i=1}^{n}S_{li}^{-1}\right)$. In the following theorem, we show that the spectral norm scales to $O(d_l^2)$ when the first-order gradients and $\lambda_l$ are bounded.

Table 1: Comparison between Exact, LiSSA, and DataInf. Computational and memory complexities are obtained for a multilayer perceptron model with $L$ layers, each with an equal number of neurons. In this case, the number of parameters in each layer is the same across different layers, and we denote it by $D \in \mathbb{N}$, *i.e.*, $d_l$ is equal to $D$ for all $l \in [L]$. DataInf has better efficiency than both Exact and LiSSA in terms of computational and memory complexities. Compared to LiSSA, DataInf leverages the closed-from expression presented in equation 5, and thus it does not require an expensive iterative algorithm.

| Method | Hessian Inversion | Underlying Approximation | Computational Complexity | Memory Complexity |
|---|---|---|---|---|
| Exact (equation 2) | Matrix inversion | | $O(nD^2L + D^3L)$ | $O(D^2)$ |
| LiSSA | Iterative update | equation 3 | $O(nD^2L)$ | $O(D^2)$ |
| DataInf (Ours) | Closed-form expression | equation 4 | $O(nDL)$ | $O(D)$ |

**Theorem 1** (Approximation error analysis). *Suppose $\max_{i \in [n]} \left\| \nabla_{\theta_l} \ell_i \right\|_\infty$ and $\lambda_l$ are bounded. Then, the spectral norm of the difference $\left\| \left( \frac{1}{n} \sum_{i=1}^n S_{li} \right)^{-1} - \frac{1}{n} \sum_{i=1}^n S_{li}^{-1} \right\|_2$ is bounded by $O(d_l^2)$.*

A proof is provided in Appendix B. Theorem 1 shows that the spectral norm is bounded by $O(d_l^2)$ when $\max_{i \in [n]} \left\| \nabla_{\theta_l} \ell_i \right\|_\infty$ and $\lambda_l$ are bounded. This assumption is generally satisfied in practice as gradients are typically bounded and we can control $\lambda_l$. One direct implication of Theorem 1 is that the total approximation error is bounded by $O(\sum_{l=1}^L d_l^2)$. This bound may be pessimistic, but the approximation error becomes more tolerable as $d_l$ is small. This is why DataInf is particularly well-suited for estimating the influence of data used for LoRA fine-tuning.

## 4 EXPERIMENTS

We investigate the empirical effectiveness of DataInf through three experiments: (i) approximation error analysis, (ii) mislabeled data detection, and (iii) influential data identification. These tasks are designed to quantitatively assess the practical efficacy of DataInf, and we also present qualitative examples in Figure 3.

**Experimental settings** For all experiments, we consider publicly available and widely used large-scale LLMs and diffusion models. We use the RoBERTa model (Liu et al., 2019) for the approximation error analysis[2] and mislabeled data detection tasks, and the Llama-2-13B-chat (Touvron et al., 2023) and the stable-diffusion-v1.5 (Rombach et al., 2022) models for the influential data identification task. During training, we use Low-Rank Adaptation (LoRA), a technique that significantly reduces the memory and computation required for fine-tuning large models (Hu et al., 2021). We fine-tune models by minimizing a negative log-likelihood function. As for the baseline influence computation methods, we consider `LiSSA` with 10 iterations (Martens, 2010; Koh & Liang, 2017), `Hessian-free` which computes a dot product of the first-order gradients, *i.e.*, $-\sum_{l=1}^L v_l^T \nabla_{\theta_l} \ell_k$ (Charpiat et al., 2019; Pruthi et al., 2020), and the proposed method `DataInf`. For all methods, we use the same damping parameter $\lambda_l = 0.1 \times (nd_l)^{-1} \sum_{i=1}^n \nabla_{\theta_l} \ell_i^T \nabla_{\theta_l} \ell_i$ following the literature (Grosse et al., 2023). We provide implementation details on datasets and hyperparameters in Appendix D.

### 4.1 APPROXIMATION ERROR ANALYSIS

Theorem 1 shows that the approximation error increases as the parameter size increases. In this experiment, we empirically study how different ranks of the LoRA matrix affect the approximation error, where we anticipate the approximation error increases as the rank increases. In addition, we compare the approximation ability between the three influence calculation methods `DataInf`, `Hessian-free`, and `LiSSA`. For each influence method, we evaluate the Pearson correlation

---

[2]Computing exact influence functions of the Llama-2-13B-chat and stable-diffusion-v1.5 models is highly expensive. Hence, we used a relatively small model, RoBERTa-large, to obtain the exact influence function value presented in equation 2. This allows us to systemically perform the approximation error analysis.

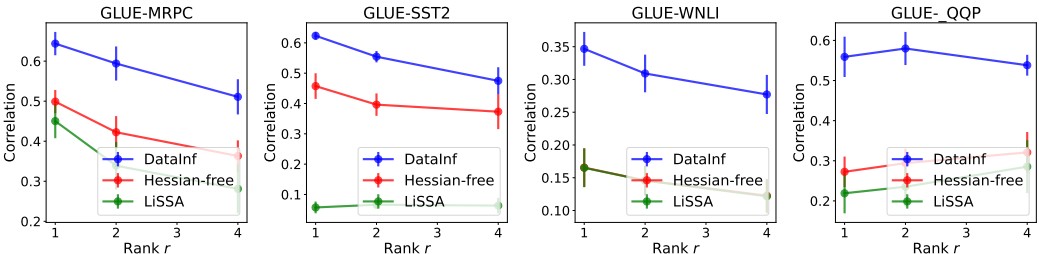

Figure 1: Correlation coefficient comparison of the three influence computation methods. The correlation coefficient captures the similarity to the exact computation of the influence function (equation 2), and thus the higher the correlation coefficient, the better. The error bar indicates a 95% confidence interval based on 20 independent runs. `DataInf` is significantly more correlated with the exact influence values than other methods for all ranks $r \in \{1, 2, 4\}$, showing better approximation ability. Also, the correlation coefficient of `DataInf` generally decreases as the rank increases, consistent with our theoretical analysis.

coefficient with the exact influence function presented in equation 2. The higher the correlation is, the better. We use the four binary classification GLUE datasets (Wang et al., 2018). To simulate the situation where a fraction of data points are noisy, we consider noisy GLUE datasets by synthetically generating mislabeled training data points. We flip a binary label for 20% of randomly selected training data points. The low rank is denoted by $r$ and is selected from $\{1, 2, 4\}$.

**Results** Figure 1 shows that `DataInf` is significantly more correlated with the exact influence method than `Hessian-free` and `LiSSA` for all ranks $r \in \{1, 2, 4\}$. For instance, when the dataset is GLUE-MRPC and $r = 1$, the correlation coefficient of `DataInf` is 0.64 while `Hessian-free` and `LiSSA` achieve 0.50 and 0.45, respectively. We observe `LiSSA` is highly unstable, leading to a worse correlation than `Hessian-free`. This instability is potentially due to the `LiSSA`'s iterative updates which often make the inverse Hessian vector product fail to converge. In addition, the correlation generally decreases as the rank increases, which aligns with our finding in Theorem 1; Its approximation error increases as the parameter size increases. Overall, `DataInf` better approximates the exact influence function values than other methods in terms of correlation coefficients. This result suggests that DataInf is well-suited for fine-tuning techniques, where the number of learnable parameters is smaller.

## 4.2 MISLABELED DATA DETECTION

Given that mislabeled data points often negatively affect the model performance, it is anticipated that their influence value should be larger than that of clean data points—when they are included, then the loss is likely to increase. In this experiment, we empirically investigate the mislabeled data detection ability of the three influence computation methods as well as the exact influence function (equation 2), which we denote by `Exact`. We consider the same noisy GLUE datasets used in the approximation error analysis. Like the previous experiment, ground-truth annotations for mislabeled data (*e.g.*, one for mislabeled data and zero for clean data) are used only to evaluate the quality of the influence function, not for fine-tuning and influence calculations.

As for the evaluation metric, we use the area under the curve (AUC) score between influence values and the binary annotations for mislabeled data to capture the quality of the influence function values. This AUC measures the probability that a score randomly selected from a class of mislabeled data is greater than that of a class of clean data. That is, an influence function likely to assign large values to mislabeled data points will have high AUC values. We measure the runtime for computing the influence function for every training data point when one NVIDIA A40 GPU processor is used. The rank $r$ of the LoRA matrix is set to be 4, but we provide additional experimental results for $r = 1$, $r = 2$, and $r = 8$ in Appendix E, where we find a consistent pattern.

**Results** Figure 2 shows that `DataInf` achieves significantly better detection ability than `Hessian-free` and `LiSSA` on all four datasets. Compared to `Exact`, `DataInf` presents overall

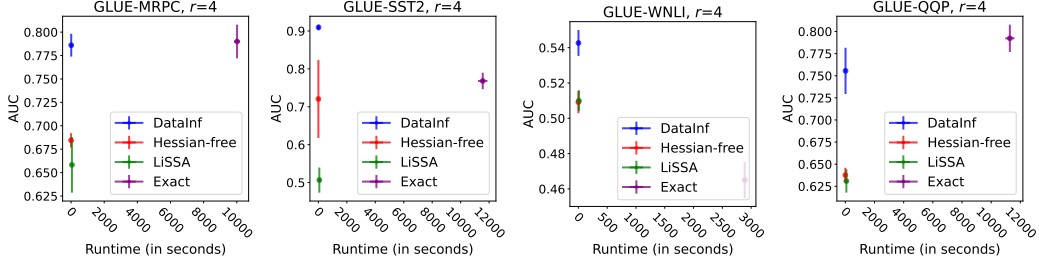

Figure 2: Mislabeled data detection ability comparison of the four influence computation methods when the rank of LoRA matrix $r$ is 4. The detection ability is evaluated with AUC, and the error bar indicates a 95% confidence interval based on 20 independent runs. `DataInf` shows better than or comparable detection ability to `Exact`, and it significantly outperforms `Hessian-free` and `LiSSA` on all four datasets. As for the runtime, `DataInf` is much faster than `Exact`, demonstrating the practical effectiveness of our method.

comparable results. Interestingly, we find that `DataInf` is sometimes better than `Exact`. This is potentially because `Exact` is not designed for detecting mislabeled data. Even correctly labeled data can have a large influence, especially when it is close to a classifier boundary, *i.e.*, highly ambiguous data. Another potential reason is that the damping parameter $\lambda_l$ can cause the degradation of `Exact`, but we leave a rigorous analysis of this as a future research topic. In terms of runtime, `DataInf` shows superior computational efficiency. For instance, on the GLUE-QQP dataset, `DataInf` takes 13 seconds while `LiSSA` and `Exact` take 70 and 11279 seconds, respectively, for computing 4500 influence function values. Across the four datasets, ours is 5.5 and 1149.6 times faster than `LiSSA` and `Exact`, respectively, on average. While `Hessian-free` is shown to be the fastest method as it does not require to compute the Hessian, its performance is strictly worse than `DataInf`.

## 4.3 INFLUENTIAL DATA IDENTIFICATION

To further illustrate the usefulness of `DataInf`, we assess how accurately it can identify influential data points in text generation and text-to-image generation tasks. We use the Llama-2-13B-chat model (Touvron et al., 2023) for the text generation task, and the stable-diffusion-v1.5 model (Rombach et al., 2022) for the text-to-image generation task. Both models are publicly available and widely used in literature.

We construct three demonstrative datasets for the text generation task: (i) Sentence transformations, (ii) Math word problems (without reasoning), and (iii) Math word problems (with reasoning). The detailed description for each task and dataset is given in Table 3 in Appendix D. Each dataset contains 10 distinct classes, with 100 total data points in each class. We partition the 100 examples into 90 training data (used for LoRA) points and 10 test data points. For text-to-image generation, we consider two tasks: (i) style generation and (ii) subject generation. For the style generation task, we combine three publicly available image-text pair datasets where each dataset represents different style: cartoons (Norod78, 2023), pixel-art (Jainr3, 2023), and line sketches (Zoheb, 2023). For each style, we use 200 training image-text pairs and 50 test image-text pairs for a total of 600 training data points and 150 test data points. For the subject generation task, we use the DreamBooth dataset (Ruiz et al., 2023). There are 31 different subjects and for each subject, 3 data points are used for the training dataset and 1 to 3 data points are used for the validation dataset. The detailed prompts are provided in Appendix D.

If some training data points help minimize a test data point's loss, then their influence function value should be negative—a validation loss should decrease when the same class data point is added. Based on this intuition, we utilize two evaluation metrics. First, for each test data point, we make a pseudo label for every training data point. This pseudo label is one if its label is the same as the test data point's label, and zero otherwise. We then compute the AUC between the negative influence function values and the pseudo labels. Ideally, a large AUC is expected because the same class will have a negative influence value. We report the average AUC across test data points. We denote this by class detection AUC. Second, for each test data point, we compute the percentage of training points with the same class as the test example among the $s$ smallest influential training points. Here, $s$ is set to

Table 2: AUC and recall comparison of `Hessian-free` and `DataInf` on influential data identification tasks. `LiSSA` is excluded in this experiment as it often fails to converge due to its instability. The average and standard deviation of the AUC and recall across test data points are denoted by "average±standard deviation", and the higher, the better for both metrics. `DataInf` significantly outperforms `Hessian-free` in both metrics across 5 different tasks.

| Task & Model | Method | Class detection (AUC) ↑ | Class detection (Recall) ↑ |
|---|---|---|---|
| Sentence transformations *Llama-2-13B-chat* | `Hessian-free` `DataInf` | $0.999 \pm 0.002$ **$1.000 \pm 0.000$** | $0.985 \pm 0.033$ **$0.997 \pm 0.010$** |
| Math problems (no reasoning) *Llama-2-13B-chat* | `Hessian-free` `DataInf` | $0.770 \pm 0.174$ **$1.000 \pm 0.000$** | $0.258 \pm 0.388$ **$0.999 \pm 0.006$** |
| Math problems (with reasoning) *Llama-2-13B-chat* | `Hessian-free` `DataInf` | $0.772 \pm 0.173$ **$1.000 \pm 0.001$** | $0.258 \pm 0.388$ **$0.996 \pm 0.025$** |
| Text-to-image style generation *stable-diffusion-v1.5* | `Hessian-free` `DataInf` | $0.692 \pm 0.007$ **$0.820 \pm 0.005$** | $0.533 \pm 0.008$ **$0.687 \pm 0.006$** |
| Text-to-image subject generation *stable-diffusion-v1.5* | `Hessian-free` `DataInf` | $0.820 \pm 0.000$ **$0.865 \pm 0.000$** | $0.210 \pm 0.003$ **$0.315 \pm 0.003$** |

be the number of training examples per class. We report the average percentage across the test data points and denote this by recall. These metrics are intended to assess how effectively each method can identify training data points that belong to the same class as a test example as more helpful than one that belongs to a different class.

**Results**  Evaluation metrics for each task and model are reported in Table 2. `DataInf` significantly outperforms `Hessian-free` on all tasks and across all metrics in identifying influential data. Of note, `LiSSA` demonstrated significant numerical instability across all four tasks and models, which produced invalid influence values in some runs. Even after gradient rescaling, we observed that `LiSSA` collapses to `Hessian-free` across all tasks. We hypothesize that this instability can be attributed to the iterative nature of `LiSSA` and the high-dimensional nature of large-scale models. Qualitative examples with test examples and the corresponding most and least influential training data points based on their absolute values are shown in Figure 3.

## 5 RELATED WORKS

Assessing the impact of individual training data points on model accuracy or predictions has been widely studied in data valuation literature. One of the most widely adopted classes of data valuation methods is based on the marginal contribution, which measures the average change in a model's performance when a specific data point is removed from a set of data points. Many standard methods belong to this category including leave-one-out and various Shapley-based methods (Ghorbani & Zou, 2019; Jia et al., 2019; Kwon & Zou, 2022; Wang & Jia, 2023; Wang et al., 2023). In addition to these methods, several alternative approaches have been proposed using reinforcement learning (Yoon et al., 2020) or out-of-bag accuracy (Kwon & Zou, 2023). Unlike all aforementioned data valuation methods, the influence function is based on gradients, conveying the rigorous and intuitive notion of data values. For an extensive review of these methods, we refer the reader to Jiang et al. (2023).

When it comes to the empirical performance on downstream machine learning tasks, other non-influence-based data contribution methods often outperform the influence function (Park et al., 2023; Jiang et al., 2023). However, the majority of previous studies have focused on relatively small models and datasets. This limitation arises from the computational infeasibility of existing algorithms, which typically require the training of numerous models to obtain reliable data values (Sim et al., 2022; Feldman & Zhang, 2020). When LLMs or diffusion models are deployed, data valuation methods that require model training are not practically applicable. While several methods capture the value of data at model initialization and do not require a training process (Nohyun et al., 2022; Wu et al., 2022), their performance is mostly examined on relatively small neural network models. The development of an efficient computational method for LLMs or diffusion models is of critical significance, and it is the main contribution of this paper.

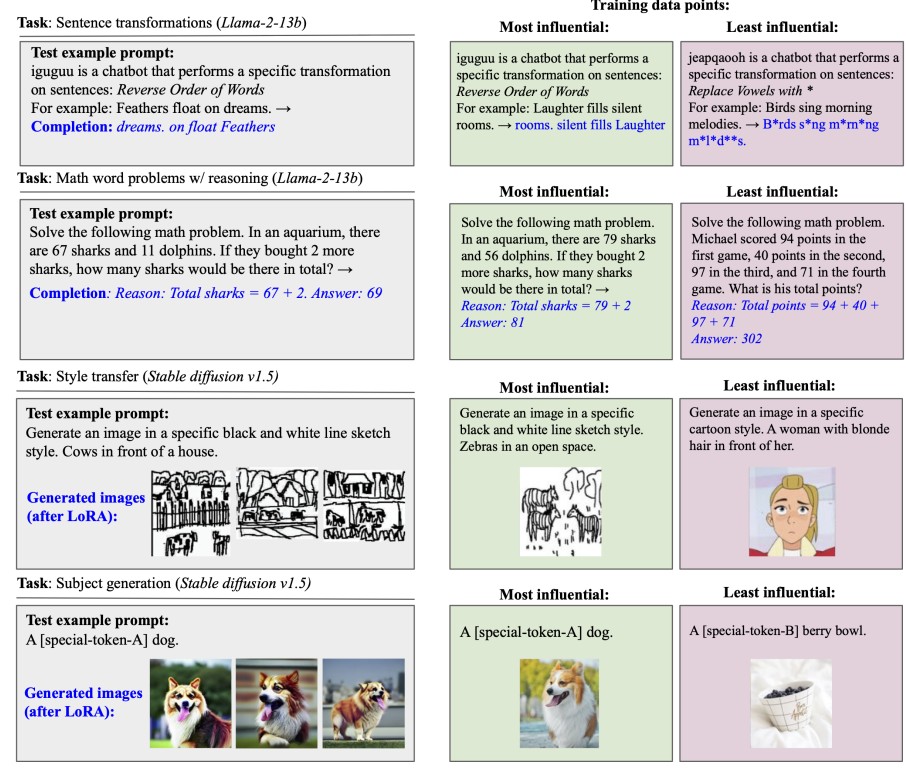

Figure 3: Illustrative examples of most and least influential training data points discovered using `DataInf` across the text generation and text-to-image generation tasks performed with the Llama-2-13B-chat and stable-diffusion-v1.5 models. The most (*resp.* least) influential data point has the largest (*resp.* smallest) absolute influence on the test example among training data points. `DataInf` has successfully identified the most influential data points, which exhibit a high degree of relevance to test example prompts. Conversely, the least influential data points identified by `DataInf` demonstrate lower relevance. In essence, `DataInf` is effective at detecting influential data points.

As a concurrent and independent work, Grosse et al. (2023) proposed the Eigenvalue-Kronecker-factored approximate curvature (EK-FAC)-based algorithm that efficiently computes the influence function. While it was applied to LLMs (though not diffusions), the EK-FAC method highly depends on the network architecture, and thus its application to LoRA-tuned models is not straightforward. The implementation of EK-FAC is also not public. We provide a more detailed explanation in Appendix C.

## 6  CONCLUSION

We propose DataInf, an efficient influence computation algorithm that is well-suited for parameter-efficient fine-tuning and can be deployed to LLMs and large diffusion models. DataInf is effective in identifying mislabeled data points and retrieving the most and least influential training data points on model generations. DataInf is orders of magnitude faster than state-of-the-art influence computation methods and is memory efficient, and thus it can be practically useful for enabling data-centric analysis of large models such as LLMs.

In the literature, there are not many quantitative evaluation metrics for the utility of influence scores. This also presents limitations for evaluating DataInf. We tried to address it by using proxies such as data points in the same class should have greater influence than data points in a different class, and mislabeled points should increase test loss. Additional downstream of the utility of influence scores for generative AI is an important direction of future work.

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

# A    PROPOSED ALGORITHM

We provide a pseudo algorithm in Algorithm 1.

---

**Algorithm 1** DataInf: Efficient influence computation

---

**Input:** A training dataset $\mathcal{D} = \{(x_1, y_1), \ldots, (x_n, y_n)\}$, a validation dataset $\mathcal{D}^{(\text{val})} = \{(x_1^{(\text{val})}, y_1^{(\text{val})}), \ldots, (x_m^{(\text{val})}, y_m^{(\text{val})})\}$, an objective function $\ell : \mathcal{Y} \times \mathcal{Y} \to \mathbb{R}$, a deep neural network model $f_\theta = f_{\theta_L} \circ \cdots \circ f_{\theta_1}$ where $\theta = \{\theta_1, \ldots, \theta_L\}$ and $\theta_l \in \mathbb{R}^{d_l}$ for $l \in [L]$.
**Output:** Influence functions $\{\mathcal{I}_{\text{DataInf}}(x_1, y_1), \ldots, \mathcal{I}_{\text{DataInf}}(x_n, y_n)\}$.

# Step 1: Compute the first-order gradients
**for** $l$ in $[L]$ **do**
    **for** $i$ in $[n]$ **do**
        Compute $\nabla_{\theta_l} \ell(y_i, f_\theta(x_i))$                            ▷ Training data points
    **end for**
    Compute $\frac{1}{m} \sum_{j=1}^{m} \nabla_{\theta_l} \ell(y_j^{(\text{val})}, f_\theta(x_j^{(\text{val})})) =: v_l$.          ▷ Validation data points
**end for**

# Step 2: Compute the inverse Hessian-vector product
**for** $l$ in $[L]$ **do**
    Compute $\lambda_l := 0.1 \times (nd_l)^{-1} \sum_{i=1}^{n} \nabla_{\theta_l} \ell(y_i, f_\theta(x_i))^T \nabla_{\theta_l} \ell(y_i, f_\theta(x_i))$
    $v_l \leftarrow 0$.
    **for** $i$ in $[n]$ **do**
        $c_{li} \leftarrow v_l^T \nabla_{\theta_l} \ell(y_i, f_\theta(x_i)) / \left( \lambda_l + \left\| \nabla_{\theta_l} \ell(y_i, f_\theta(x_i)) \right\|_2^2 \right)$
                                                       ▷ A normalizing constant
        $r_l \leftarrow r_l + (n\lambda_l)^{-1} \left( v_l - c_{li} \nabla_{\theta_l} \ell(y_i, f_\theta(x_i)) \right)$.
    **end for**
**end for**

# Step 3: Compute the influence function
**for** $i$ in $[n]$ **do**
    $\mathcal{I}_{\text{DataInf}}(x_k, y_k) \leftarrow -\sum_{l \in [L]} r_l^T \nabla_{\theta_l} \ell(y_k, f_\theta(x_k))$.
**end for**

---

# B    PROOF OF THEOREM 1

We provide a proof of Theorem 1 below.

*Proof.* We set $S_{li} := \nabla_{\theta_l} \ell_i \nabla_{\theta_l} \ell_i^T + \lambda_l I_{d_l}$ and $\bar{S}_l := n^{-1} \sum_{i=1}^{n} S_{li}$. Then, a Taylor expansion[3] gives

$$\frac{1}{n} \sum_{i=1}^{n} S_{li}^{-1} - \left( \frac{1}{n} \sum_{i=1}^{n} S_{li} \right)^{-1} = \frac{1}{n} \sum_{i=1}^{n} \left( S_{li} - \bar{S}_l \right)^T \left( \int_0^1 (tS_{li} + (1-t)\bar{S}_l)^{-3} 2(1-t) dt \right) \left( S_{li} - \bar{S}_l \right)$$

$$\preceq \frac{2}{\lambda_l^3} \frac{1}{n} \sum_{i=1}^{n} \left( S_{li} - \bar{S}_l \right) \left( S_{li} - \bar{S}_l \right)^T.$$

The inequality is due to the maximum eigenvalue of $(tS_{li} + (1-t)\bar{S}_l)^{-3}$ is upper bounded by $\lambda_l^{-3}$ for $0 < t < 1$. Then, the spectral norm of this matrix is upper bounded as follows:

$$\left\| \frac{1}{\lambda_l^3} \frac{1}{n} \sum_{i=1}^{n} \left( S_{li} - \bar{S}_l \right) \left( S_{li} - \bar{S}_l \right)^T \right\|_2 \leq \frac{2}{\lambda_l^3} \frac{1}{n} \sum_{i=1}^{n} \left\| S_{li} - \bar{S}_l \right\|_2^2$$

---

[3]This Taylor expansion is presented in (Bach, 2022).

$$\leq \frac{2}{\lambda_l^3} \frac{1}{n} \sum_{i=1}^n \left( \|S_{li}\|_2 + \|\bar{S}_l\|_2 \right)^2 .$$

The first inequality is due to the triangle inequality and the second inequality is from $\lambda_{\max}(A - B) \leq \lambda_{\max}(A) + \lambda_{\max}(B)$ for real symmetric matrices $A$ and $B$. Here, $\lambda_{\max}(A)$ denotes the largest eigenvalue of a matrix $A$.

For all $l \in [L]$ and $i \in [n]$, since $S_{li}$ is a semi-positive definite matrix, we have $\|S_{li}\|_2 = \lambda_{\max}(S_{li}) \leq \mathrm{tr}(S_{li})$ and $\|\bar{S}_l\|_2 = \lambda_{\max}(\bar{S}_l) \leq \mathrm{tr}(\bar{S}_l)$. Hence,

$$\frac{2}{\lambda_l^3} \frac{1}{n} \sum_{i=1}^n \left( \|S_{li}\|_2 + \|\bar{S}_l\|_2 \right)^2 \leq \frac{2}{\lambda_l^3} \frac{1}{n} \sum_{i=1}^n \left( \mathrm{tr}\left(S_{li}\right) + \mathrm{tr}\left(\bar{S}_l\right) \right)^2 .$$

Since $\|\nabla_{\theta_l} \ell_i\|_\infty$ is upper bounded by a constant, there exists a constant $M > 0$ such that $\mathrm{tr}\left(S_{li}\right) + \mathrm{tr}\left(\bar{S}_l\right) \leq M d_l$. This implies that:

$$\frac{2}{\lambda_l^3} \frac{1}{n} \sum_{i=1}^n \left( \|S_{li}\|_2 + \|\bar{S}_l\|_2 \right)^2 \leq \frac{2M^2 d_l^2}{\lambda_l^3} = O(d_l^2).$$

$\square$

## C  EXISTING METHODS

In this section, we provide three existing influence computation methods: EK-FAC and retraining-based methods.

**EK-FAC**  To this end, we suppose $f_\theta$ is a multilayer perceptron model. For $l \in [L]$, we denote the $l$-th layer output by $h_l := f_{\theta_l} \circ \cdots \circ f_{\theta_1}$ and the associated pre-activated output by $g_l$. Then, for all $k \in [n]$, we have $\nabla_{\theta_l} \ell_k = h_{l-1}(x_k) \otimes \nabla_{g_l} \ell_k$ where $\otimes$ denotes the Kronecker product. Moreover, the Hessian for $\theta_l$ is given as follows:

$$G_l(\theta^*) = \frac{1}{n} \sum_{i=1}^n \left( h_{l-1}(x_i) \otimes \nabla_{g_l} \ell_i \right) \left( h_{l-1}(x_i) \otimes \nabla_{g_l} \ell_i \right)^T$$

$$= \frac{1}{n} \sum_{i=1}^n \left( h_{l-1}(x_i) h_{l-1}(x_i)^T \right) \otimes \left( \nabla_{g_l} \ell_i \nabla_{g_l} \ell_i^T \right)$$

The second equality is due to the mixed-product property of the Kronecker product. George et al. (2018) proposed to approximate $G_l(\theta^*)$ with the following equation:

$$G_l(\theta^*) \approx \frac{1}{n} \sum_{i=1}^n h_{l-1}(x_i) h_{l-1}(x_i)^T \otimes \frac{1}{n} \sum_{i=1}^n \nabla_{g_l} \ell_i \nabla_{g_l} \ell_i^T =: A_{l-1} \otimes B_l$$

This approximation is accurate when one of the following approximations holds:

$$\nabla_{g_l} \ell_i \nabla_{g_l} \ell_i^T \approx \frac{1}{n} \sum_{j=1}^n \nabla_{g_l} \ell_j \nabla_{g_l} \ell_j^T \quad \forall i \in [n], \tag{6}$$

or

$$h_{l-1}(x_i) h_{l-1}(x_i)^T \approx \frac{1}{n} \sum_{j=1}^n h_{l-1}(x_j) h_{l-1}(x_j)^T \quad \forall i \in [n]. \tag{7}$$

The assumptions in equation 6 and equation 7 essentially assume that $h_{l-1}$ and $\nabla_{g_l} \ell$ are constant across different training data points. While there is no clear intuition, the approximation leads to a computationally efficient form. Specifically, we let $A_{l-1} = Q_{A_{l-1}} \Lambda_{A_{l-1}} Q_{A_{l-1}}^T$ where $Q_{A_{l-1}}$ is an orthonormal matrix and $\Lambda_{A_{l-1}}$ is a diagonal matrix obtained by eigenvalue decomposition. Similarly, we factorize $B_l = Q_{B_l} \Lambda_{B_l} Q_{B_l}^T$. Then, we have:

$$(G_l(\theta^*) + \lambda_l I_{d_l})^{-1} = (Q_{A_{l-1}} \otimes Q_{B_l})(\Lambda_{A_{l-1}} \otimes \Lambda_{B_l} + \lambda_l I_{d_l})^{-1}(Q_{A_{l-1}} \otimes Q_{B_l})^T$$

That is, $(G_l(\theta^*) + \lambda_l I_{d_l})^{-1}$ can be obtained by applying eigenvalue decomposition of $A_{l-1}$ and $B_l$. Compared to naive matrix inversion $(G_l(\theta^*) + \lambda_l I_{d_l})^{-1}$, this approximation method has computational efficiency as the size of $A_{l-1}$ and $B_l$ is much smaller than $d_l$. This method is called Kronecker Factorization, also known as KFAC. George et al. (2018) showed that that using $\Lambda_{A_{l-1}} \otimes \Lambda_{B_l}$ can be suboptimal in approximating with a diagonal matrix $\Lambda$ where the $i$-th diagonal element is set to be $\Lambda_{ii} = n^{-1} \sum_{j=1}^{n} ((Q_{A_{l-1}} \otimes Q_{B_l}) \nabla_{\theta_l} \ell_j)_i^2$. A naive computation of the left-hand side requires $O(d_l^3)$ operations, but both KFAC and EK-FAC can be done in $O(d_l^{3/2})$ operations when the size of both $A_{l-1}$ and $B_l$ is $\sqrt{d_l}$. Hence, the total computational complexity is $O(\sum_{l=1}^{L} (nd_l + d_l^{3/2}))$ with $O(\max_{l \in [L]} d_l^2)$ memory complexity.

EK-FAC has a computational advantage over a naive version of LiSSA, but it might not be applicable to general deep neural network models as it highly depends on the model architecture—the gradient should be expressed as $\nabla_{\theta_l} \ell_k = h_{l-1}(x_k) \otimes \nabla_{g_l} \ell_k$, which it might not be true for LoRA fine-tuned models or transformer-based models. This method has been used in an independent and concurrent work (Grosse et al., 2023).

**Retraining-based method** For $M \in [n]$, we denote a set of subsets with the same cardinality by $\mathcal{S}^{(M)} := \{S : S \subseteq \mathcal{D}, |S| = M\}$. For $i \in [n]$, we set $\mathcal{S}_{i,\text{in}}^{(M)} := \{S \in \mathcal{S}^{(M)} : (x_i, y_i) \in S\}$ and $\mathcal{S}_{i,\text{out}}^{(M)} := \{S \in \mathcal{S}^{(M)} : (x_i, y_i) \notin S\}$. Note that $\mathcal{S}_{i,\text{in}}^{(M)} \cup \mathcal{S}_{i,\text{out}}^{(M)} = \mathcal{S}$ and $\mathcal{S}_{i,\text{in}}^{(M)} \cap \mathcal{S}_{i,\text{out}}^{(M)} = \{\}$ for all $i \in [n]$. Feldman & Zhang (2020) proposed to compute the influence of a data point $(x_i, y_i) \in \mathcal{D}$ on a loss value $\ell(y^*, f_{\theta^*}(x^*))$ as a difference of model outputs.

$$\frac{1}{|\mathcal{S}_{i,\text{in}}^{(M)}|} \sum_{S \in \mathcal{S}_{i,\text{in}}^{(M)}} \ell(y^*, f_{\theta_S}(x^*)) - \frac{1}{|\mathcal{S}_{i,\text{out}}^{(M)}|} \sum_{S \in \mathcal{S}_{i,\text{out}}^{(M)}} \ell(y^*, f_{\theta_S}(x^*)),$$

where

$$\theta_S := \text{argmin}_{\theta \in \Theta} \frac{1}{|S|} \sum_{i \in S} \ell(y_i, f_\theta(x_i)).$$

This retraining-based method is flexible in that any deep neural network model can be used for $f_\theta$, however, it is extremely challenging to compute because it necessitates the training of numerous models.

# D   IMPLEMENTATION DETAILS

In this section, we provide implementation details on datasets, models, and loss functions we used in Section 4. Python-based implementation codes are available at `https://github.com/ykwon0407/DataInf`.

## D.1   APPROXIMATION ERROR ANALYSIS AND MISLABELED DATA DETECTION

**Datasets** We consider the four binary classification GLUE benchmarking datasets (Wang et al., 2018). Namely, the four datasets are GLUE-MARPC (Dolan & Brockett, 2005, paraphrase detection), GLUE-SST2 (Socher et al., 2013, sentiment analysis), GLUE-WNLI (Levesque et al., 2012, inference), and GLUE-QQP (question-answering)[4]. We used the training and validation splits of the dataset available at HuggingFace Datasets library (Lhoest et al., 2021). Only the training dataset is used to fine-tune the model, and we compute the influence of individual training data points on the validation loss. For GLUE-SST2 and GLUE-QQP, we randomly sample 4500 (*resp.* 500) samples from the original training (*resp.* validation) dataset.

**Model** We use LoRA to fine-tune the RoBERTa-large model, a 355M parameter LLM trained on the large-scale publicly available natural language datasets (Liu et al., 2019). We apply LoRA to every value matrix of the attention layers of the RoBERTa-large model. We downloaded the pre-trained model from the HuggingFace Transformers library (Wolf et al., 2020). Across all fine-tuning runs,

---

[4]`https://quoradata.quora.com/First-Quora-Dataset-Release-Question-Pairs`

we use a learning rate of $3 \times 10^{-4}$ with a batch size of 32 across 10 training epochs. As for the LoRA hyperparameters, the dropout rate is set to be $0.05$. We choose the rank of the LoRA matrix $r$ from $\{1, 2, 4, 8\}$ and $\alpha$ is always set to be $r$. The training was performed on a single machine with one NVIDIA A40 GPU using the HuggingFace Peft library (Mangrulkar et al., 2022).

For the `Exact` method, we exclude the case $r = 8$ to compute it in a reasonable time. Specifically, with one NVIDIA A40 GPU processor and the GLUE-MRPC dataset, it takes more than 18 hours when $r = 8$. Also, we find that the LoRA with a smaller $r$ can yield a similar model performance to the LoRA with $r = 8$. See Figure 6.

**Loss** All GLUE benchmarking datasets we used are binary classification datasets, so we used a negative log-likelihood function as a loss function. For a sequence of input tokens $x = (x_1, \ldots, x_T)$ and its label $y$, we consider $\ell(y, f_\theta(x)) = -\log p(y \mid f_\theta(x))$ where $f_\theta$ is a composition model consists of the RoBERTa-large model to convert text data into numerical embeddings, and a logistic model to perform classification on the embedding space. We set $T = 128$.

### D.2 INFLUENTIAL DATA IDENTIFICATION

#### D.2.1 TEXT GENERATION

**Datasets** Full dataset prompts are described in Tables 5 and 6.

**Model** We use LoRA to fine-tune Llama-2-13B-chat, an LLM that has 13 billion parameters, is pre-trained on 2 trillion tokens, and is optimized for dialogue use cases (Touvron et al., 2023). We apply LoRA to every query and value matrix of the attention layer in the Llama-2-13B-chat model. Across all fine-tuning runs, we use a learning rate of $3 \times 10^{-4}$, LoRA hyperparameters $r = 8$ and $\alpha = 32$, in 8-bit quantization, with a batch size of 64 across 25 training epochs. The training was performed on a single machine with 4 NVIDIA V100 GPUs using the HuggingFace Peft library (Mangrulkar et al., 2022).

**Loss** We used a negative log-likelihood of a generated response as a loss function. For a sequence of input tokens $x = (x_1, \ldots, x_{T_1})$ and the corresponding sequence of target tokens $y = (y_1, \ldots, y_{T_2})$, suppose the Llama-2-13B generates a sequence of output tokens $f_\theta(x) = (f_\theta(x)_1, \ldots, f_\theta(x)_{T_2})$. $f_\theta(x)$ has the same size of $T_2$ and is generated in an auto-regressive manner. We set $T_1 = T_2 = 512$. Then, the loss function is $\ell(y, f_\theta(x)) = -\sum_{t=1}^{T_2} \log p(y_t \mid f_\theta(x)_1, \ldots, f_\theta(x)_{t-1})$.

**Experiments** The test accuracy for the base (non-fine-tuned) and fine-tuned models is shown in Table 4. We observe substantial improvements across the three tasks, with additional improvements from introducing an intermediate reasoning step to the math word problems.

#### D.2.2 TEXT-TO-IMAGE GENERATION - STYLE GENERATION

**Datasets** Figure 4 illustrates examples of images used in the text-to-image generation task. When we fine-tuned a model, a style description is added to a text sequence to instruct a style. For instance, "Generate an image in a specific {custom} style. {text-data}", where {custom} is substituted with either "cartoon", "pixelart", or "black and white line sketch", and {text-data} is substituted with a text sequence in the training dataset. We provide a detailed description in Table 7.

**Model** We also use LoRA to fine-tune stable-diffusion-v1.5 (Rombach et al., 2022). We apply LoRA to every attention layer in the stable-diffusion-v1.5 model. Across all fine-tuning runs, we use a learning rate of $10^{-4}$, LoRA hyperparameters $r = 1$ and $\alpha = 1$. We fine-tune a model with a batch size of 4 across 10000 training steps. The training was performed on a single machine with 4 NVIDIA V100 GPUs using the HuggingFace Peft library (Mangrulkar et al., 2022).

**Loss** Similar to other experiments, we used a negative log-likelihood of a generated image as a loss function. For a sequence of input tokens $x = (x_1, \ldots, x_T)$ and the corresponding target image $y$, we can compute a negative log-likelihood $\ell(y, f_\theta(x)) = -\log p(y \mid f_\theta(x))$. We set $T = 77$.

**Experiment**   We compare generated images before and after the LoRA fine-tuning. For quantitative comparison, the baseline and fine-tuned models are evaluated by comparing the Fréchet inception distance (FID) between images in the test set with images generated using the paired texts from the test set. FID for the before and after fine-tuning models is shown in Table 4.

### D.2.3   TEXT-TO-IMAGE GENERATION - SUBJECT GENERATION

We used the same setting with the text-to-image generation style generation task, but the rank of LoRA matrices is set to $4$. We here explain the datasets.

**Datasets**   Figure 4 illustrates examples of images used in the text-to-image generation task. Our models use Google's Dreambooth dataset Ruiz et al. (2023), which contains 31 unique subjects in categories like backpack, dog, bowl, and sneaker. Each subject has four to six total examples, and we take three from each subject as the training set, with the remaining as the validation. We add a unique random string to each subject to prompt the model to produce each subject. For example, to differentiate two different dogs, we use prompts "a 5a2PZ dog" and "a POVRB dog" in the training and test set.

Table 3: Description of each task and dataset used in Section 4.3. For the text generation task, we describe the full dataset prompts in Table 5.

| Task | Description |
|---|---|
| Sentence transformations | The model is asked to perform a specific transformation on a sentence. Ten different types of sentence transformations are used. To aid the model in learning distinct transformations, "chatbot" name identifiers that are uniquely associated with each transformation are added to the prompts. |
| Math problems (without reasoning) | The model is asked to provide a direct answer to a simple arithmetic word problem. Ten different types of math word problems are constructed, and random positive integers are used to construct unique data points. |
| Math problems (with reasoning) | Using the same questions with as above (Math problems without reasoning), the prompt includes an intermediate reasoning step before arriving at the final answer. |
| Text-to-image style generation | The model is asked to generate an image in a given style. Three different styles of images are used in our dataset. We illustrate examples of styles in Figure 4. |
| Text-to-image subject generation | The model is asked to generate a specific subject (e.g. a dog or a candle). We illustrate examples of styles in Figure 4. |

Table 4: Performance improvements from model fine-tuning. For the text generation tasks, we evaluate the classification accuracy. For the text-to-image generation task, we evaluate FID.

| Task | Method | Accuracy ↑ | FID ↓ |
|---|---|---|---|
| Sentence transformations | Base model | 0.01 | - |
| | Fine-tuned model | 0.35 | - |
| Math problems (no reasoning) | Base model | 0.07 | - |
| | Fine-tuned model | 0.20 | - |
| Math problems (with reasoning) | Base model | 0.08 | - |
| | Fine-tuned model | 0.31 | - |
| Text-to-image style generation | Base model | - | 243.5 |
| | Fine-tuned model | - | 189.6 |
| Text-to-image subject generation | Base model | - | 269.7 |
| | Fine-tuned model | - | 247.5 |

## E   ADDITIONAL EXPERIMENTAL RESULTS

**Approximation error analysis.**   We provide an additional approximation error analysis result when data are clean. Figure 5 shows that DataInf is more correlated with the exact influence function than

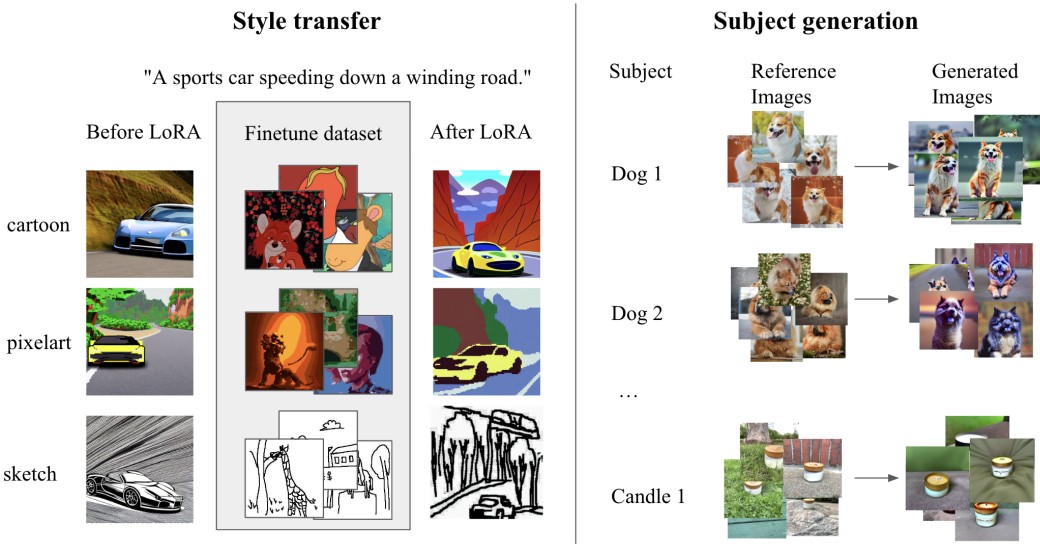

Figure 4: Examples of images used in the text-to-image generation task, along with before and after images from the LoRA fine-tuning of the stable-diffusion-v1.5 model.

Table 5: Description of the **sentence transformations task** templates. We consider 10 different types of sentence transformations. For each sentence transformation, unique identifying "chatbot" names were additionally prepended to the task prompt to assist the model in training.

| Sentence transformations | Example transformation of *"Sunrises herald hopeful tomorrows"*: |
|---|---|
| Reverse Order of Words | tomorrows. hopeful herald Sunrises |
| Capitalize Every Other Letter | sUnRiSeS hErAlD hOpEfUl tOmOrRoWs. |
| Insert Number 1 Between Every Word | Sunrises 1herald 1hopeful 1tomorrows. |
| Replace Vowels with * | S*nr*s*s h*r*ld h*p*f*l t*m*rr*ws. |
| Double Every Consonant | SSunrriisseess hheralld hhopeffull ttomorrows. |
| Capitalize Every Word | Sunrises Herald Hopeful Tomorrows. |
| Remove All Vowels | Snrss hrld hpfl tmrrws. |
| Add 'ly' To End of Each Word | Sunrisesly heraldly hopefully tomorrows.ly |
| Remove All Consonants | uie ea oeu ooo. |
| Repeat Each Word Twice | Sunrises Sunrises herald herald hopeful hopeful tomorrows. tomorrows. |

Hessian-free and LiSSA methods throughout different datasets and ranks. It shows that our finding is consistent regardless of the presence of noisy data.

**Mislabeled data detection task.** We provide additional mislabeled data detection task results when the rank $r$ is selected from $\{1, 2, 8\}$. We exclude `Exact` when $r = 8$ because it exceeds the 12-hour computation limit for all datasets. In this case, `DataInf` takes less than 25 seconds. Similar to the case $r = 4$, `DataInf` shows competitive mislabeled detection performance over other methods while achieving a short runtime.

Table 6: Description of the **math problem task** templates. We consider 10 different types of math word problems.

| Math Word Problems | Template prompt question |
|---|---|
| Remaining pizza slices | Lisa ate A slices of pizza and her brother ate B slices from a pizza that originally had C slices. How many slices of the pizza are left? *Reason:* Combined slices eaten = A + B. Left = C - (A + B). |
| Chaperones needed for trip | For every A students going on a field trip, there are B adults needed as chaperones. If C students are attending, how many adults are needed? *Reason:* Adults needed = (B * C) // A. |
| Total number after purchase | In an aquarium, there are A sharks and B dolphins. If they bought C more sharks, how many sharks would be there in total? *Reason:* Total sharks = A + C. |
| Total game points | Michael scored A points in the first game, B points in the second, C in the third, and D in the fourth game. What is his total points? *Reason:* Total points = A + B + C + D. |
| Total reading hours | Emily reads for A hours each day. How many hours does she read in total in B days? *Reason:* Total hours read = A * B. |
| Shirt cost after discount | A shirt costs A. There's a B-dollar off sale. How much does the shirt cost after the discount? *Reason:* Cost after discount = A - B. |
| Area of a garden | A rectangular garden has a length of A meters and a width of B meters. What is its area? *Reason:* Area = A * B. |
| Total savings | If Jake saves A each week, how much will he save after B weeks? *Reason:* Total savings = A * B. |
| Number of cupcake boxes | A bakery sells cupcakes in boxes of A. If they have B cupcakes, how many boxes can they fill? *Reason:* Boxes filled = B // A. |
| Interest earned | John invests A at an annual interest rate of B%. How much interest will he earn after C years? *Reason:* Interest = (A * B * C) // 100. |

Table 7: Description of the **text-to-image generation task** templates. Each style has 200 training image-text pairs and 150 validation image-text pairs.

| Image style | Text prompt |
|---|---|
| Cartoon | Generate an image in a specific cartoon style. {A text sequence of the original dataset which describes an image}. |
| Pixel Art | Generate an image in a specific pixelart style. {A text sequence of the original dataset which describes an image}. |
| Sketch | Generate an image in a specific black and white line sketch style. {A text sequence of the original dataset which describes an image}. |

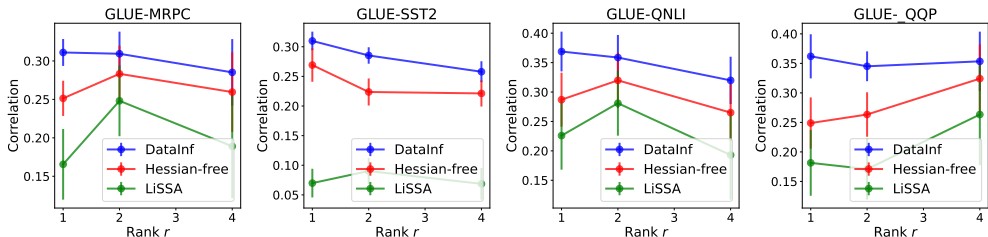

Figure 5: Correlation coefficient comparison of the three influence computation methods **when data are clean**. The experimental settings are exactly the same as the one in Figure 1 except for the presence of noisy data.

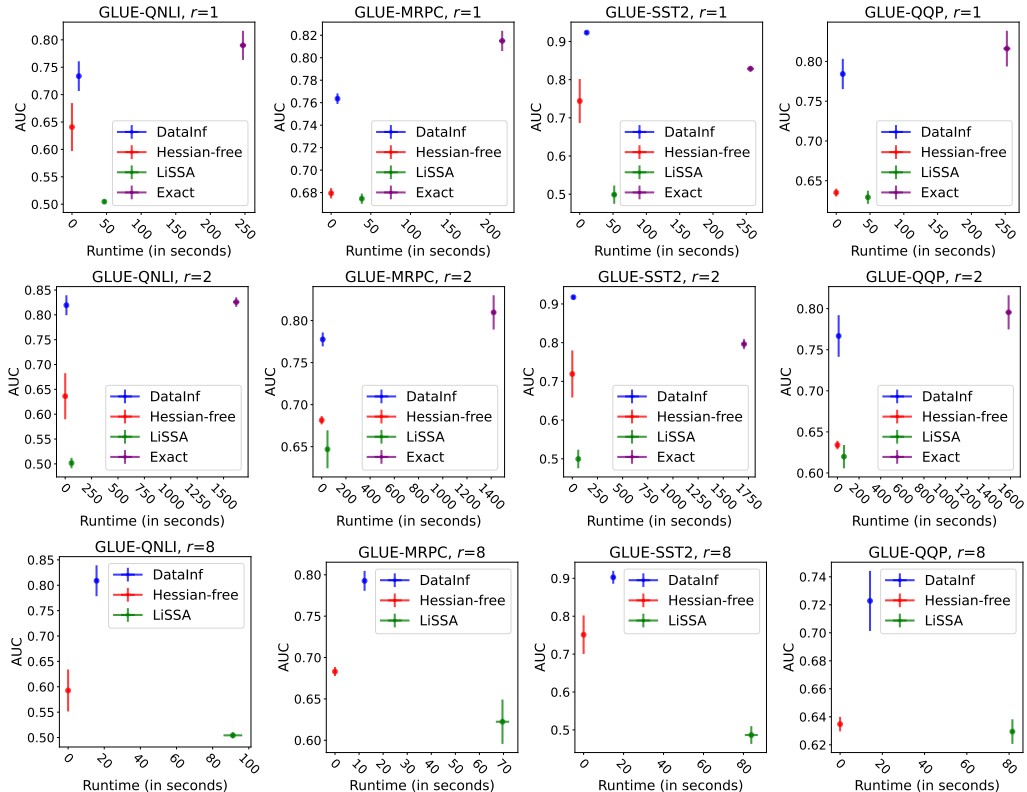

Figure 6: Mislabeled data detection task ability comparison of the four influence computation methods when the rank of LoRA matrix $r$ is (top) $1$, (middle) $2$, and (bottom) $8$. The error bar indicates a 95% confidence interval based on 20 independent runs. Similar to Figure 2, `DataInf` shows superior detection ability while achieving competitive computational efficiency.

# F  DATA SELECTION TASK

One desired property of data contribution methods is to find a representative subset that can yield a high model performance when a model is trained on that subset. To assess this ability of DataInf, we conduct data selection experiments. The experimental setting is exactly the same as the one used in Figure 2. Once the influence function is computed, we select the top 70% most beneficial data points, retrain a model from scratch with the selected subset, and evaluate the model accuracy on the holdout test dataset. We compare `DataInf` with `Hessian-free` and `LiSSA` methods. In addition, we consider two additional baseline methods: the random selection (denoted by `Random`) and the entire dataset (denoted by `Full`). We anticipate `Full` should be better than `Random` as it uses more samples.

Figure 7 shows the accuracy trajectories in the first 10 epochs. First, `DataInf` uniformly performs better than existing methods for most of the datasets across all training epochs. Also, it better performs than `Full` on most of the datasets, and we believe this is because 20% of the original datasets are mislabeled. `DataInf` detects these low-quality data points, leading to a better performance than `Full`. Furthermore, another interesting observation is that `DataInf` usually achieves the best performance in the first one to three epochs, meaning that it selectively finds a good subset that is representative of the training dataset and helps accelerate the model training.

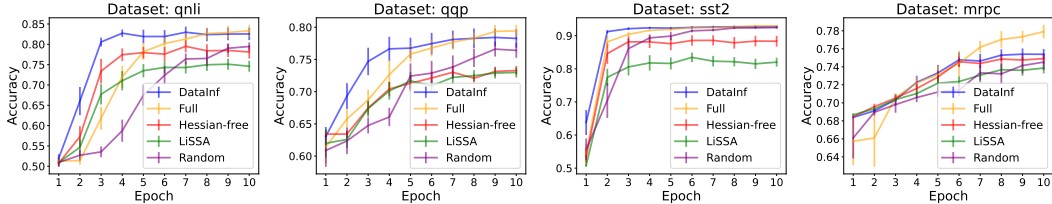

Figure 7: Data selection ability comparison of the different influence computation methods. The experimental setting is exactly the same as the one used in Figure 2. The detection ability is evaluated with classification accuracy, and the error bar indicates a 95% confidence interval based on 20 independent runs. `DataInf` significantly outperforms `Hessian-free` and `LiSSA` on all four datasets, and it is even better than `Full` except the mrpc dataset. Furthermore, `DataInf` achieves the best accuracy in the first few epochs, demonstrating that a selected subset is easy to learn and representative, which helps accelerate the model training. It shows the practical effectiveness of `DataInf` in data selection tasks, especially when a fraction of the training dataset is low-quality.

