# OpenReview forum: "DataInf: Efficiently Estimating Data Influence in LoRA-tuned LLMs and Diffusion Models"
_ICLR.cc/2024/Conference — ICLR 2024 poster_

### Official Review · Reviewer_kkF9 · 2023-10-31

**Soundness:** 3 good
**Presentation:** 3 good
**Contribution:** 3 good
**Rating:** 6
**Confidence:** 3

**Summary:**

In this paper, the authors provide a new method to estimate influence functions for large-scale generative models. Their theoretical results show that their method is well suited for LoRA-based fine-tuning settings. their compare their method in terms of approximation error, mislabeled data detection, and influential data identification. They observe improvement over existing methods in influence approximation.

**Strengths:**

+ the idea if DataInf for approximating influence function is interesting.
+ method is more efficient that existing work
+ high-level of correlation between exact influence and approximated one shows the effectiveness of their approach.

**Weaknesses:**

+ Evaluation setup for mislabeled data detection is not well-defined. I need more details to understand this experiment.
+ One of the main applications of influence functions is to find the most influential samples whose removal would significantly change model's behavior in inference time. The way this paper evaluates the most influential samples is not appropriate. In fact, we do expect influential samples to come from the same label but that is not enough. I would want to see if the most influential samples are actually influential in the generation process, e.g., model cannot generate that test image/text if those samples are removed in the fine-tuning process.
A comparison of the effect of removing top $k$ detected influential samples with different methods on the quality of generation for a given test sample is needed.

**Questions:**

There is a recent line of work that casts doubt on the usefulness of approximating influence functions. They argue that for image generation tasks, utilizing an off-the-shelf visual encoder can be more helpful in identifying influential samples. I would expect to see a comparison between the proposed method and those kinds of methods [1, 2].

In fact, what is the benefit of this approach in detecting influential samples compared to those kind of methods if proposed method doesn't bring any improvement?

Wang, Sheng-Yu, et al. "Evaluating Data Attribution for Text-to-Image Models." arXiv preprint arXiv:2306.09345 (2023).
Yang, Jiaxi, et al. "Matching-based Data Valuation for Generative Model." arXiv preprint arXiv:2304.10701 (2023).

---

> ### Author Response · Authors · 2023-11-19
> **Response to Reviewer kkF9**
>
> Thank you very much for your feedback!
>
> > Evaluation setup for mislabeled data detection is not well-defined. I need more details to understand this experiment.
>
> Thank you for this comment. As for the evaluation metric, we used the area under the curve (AUC) score between influence values and annotations for mislabeled data. To be more specific, suppose we have $n$ training data points. We take as input (i) a $n$-dimensional vector whose element represents its influence function value and (ii) another $n$-dimensional binary vector whose element represents whether or not the corresponding data point is mislabeled. If the $i$-th data point is mislabeled, then the $i$-th element of the annotation vector is one, zero otherwise. We regard the annotation vector as binary labels and the influence function as scores for this annotation. In this context, AUC captures the quality of the influence function values as it measures the probability that a score randomly selected from a positive class is greater than that of a negative class. That is, an influence function that assigns large values to mislabeled data points will have high AUC values. We have clarified the evaluation setup in the revision.
>
> > There is a recent line of work that casts doubt on the usefulness of approximating influence functions. They argue that for image generation tasks, utilizing an off-the-shelf visual encoder can be more helpful in identifying influential samples. I would expect to see a comparison between the proposed method and those kinds of methods [1, 2].
>
> Thank you for this comment. Data contribution estimation for general deep neural network models, not just image generation models, is an important open question, so we would like to highlight that our main focus is to propose a practical approximation for the influence function that can be applied to any deep neural network model. As a result, our method is designed to be more general. Moreover, [2] has not been published in a conference yet.
>
> As the reviewer mentioned, existing influence function methods have been recognized as less useful, and we believe it is because the approximation is inaccurate. It has been actually observed in our work as well: Figure 1 demonstrates that existing methods are less accurate than DataInf, and Figure 2 illustrates DataInf outperforms them in downstream tasks.

---

> ### Author Response · Authors · 2023-11-20
> **Response to Reviewer kkF9 - 2**
>
> > One of the main applications of influence functions is to find the most influential samples whose removal would significantly change model's behavior in inference time. The way this paper evaluates the most influential samples is not appropriate. In fact, we do expect influential samples to come from the same label but that is not enough. I would want to see if the most influential samples are actually influential in the generation process, e.g., model cannot generate that test image/text if those samples are removed in the fine-tuning process. A comparison of the effect of removing top $k$ detected influential samples with different methods on the quality of generation for a given test sample is needed.
>
> We greatly appreciate the reviewer for these thought-provoking comments. While we have systematically evaluated the quality of the influence function methods through approximation error analysis and mislabeled data detection tasks with repetitions, we agree with the reviewer that a good influence method is desired to perform well in the data selection task. To address your comment, we have added the data selection experiment in Appendix F (the last page of the revision)
>
> As a brief summary, we find that DataInf is very effective in data selection, demonstrating that it yields better model performance than existing influence function methods. In particular, a subset selected by DataInf often yields a better model than that based on the full dataset when the dataset is noisy. Furthermore, we find that a subset selected by DataInf achieves the best model performance in the first few epochs and is able to accelerate the model training. We believe DataInf selectively finds a good subset by excluding ambiguous or mislabeled data points, and hence it can help accelerate model training.
>
> Thank you very much again for your time and constructive feedback! We hope you would consider increasing your score if we have answered your questions. Please let us know if you have additional questions and we are happy to follow up. Thanks!

---

> > ### Comment · Reviewer_kkF9 · 2023-11-21
> >
> > Thanks for detailed explanations. I changed the score to 6.

---

> > > ### Author Response · Authors · 2023-11-22
> > > **Thank you!**
> > >
> > > We greatly thank the reviewer for your very constructive feedback and re-evaluation based on our rebuttal.

---

### Official Review · Reviewer_oMqQ · 2023-11-01

**Soundness:** 3 good
**Presentation:** 2 fair
**Contribution:** 3 good
**Rating:** 6
**Confidence:** 2

**Summary:**

In this paper, the author propose an efficient for the influence function. Different from LoRA, the author provide another efficient way to train the large language model.

**Strengths:**

DataInf has superior efficiency in terms of computational and memory complexities compared to other methods such as LiSSA.
The method can be applied on some popular framework such as LoRA.

**Weaknesses:**

While DataInf is efficient, it uses an approximation that is not always equal to the exact computation. This could lead to significant errors in certain cases.

**Questions:**

Are there any specific model configurations where you would advise not using DataInf?

---

> ### Author Response · Authors · 2023-11-19
> **Response to Reviewer oMqQ**
>
> Thank you very much for your feedback!
>
> > While DataInf is efficient, it uses an approximation that is not always equal to the exact computation. This could lead to significant errors in certain cases.
>
> Thank you for this comment. We want to highlight that almost all of the methods for computing influence function and data value more generally require approximation due to the fundamental difficulties in estimating these values. Therefore, making approximations is necessary and not a reason to reject this work. Our experiments, supported by Theorem 1, show that the approximation error of DataInf is relatively small for LoRA fine-tuned models. Our empirical results also show that DataInf is both computationally efficient and achieves accurate approximations. Given that there is currently no exact method for computing influence scores for LLMs, we believe the efficiency and performance of DataInf make it a useful contribution.
>
> > Are there any specific model configurations where you would advise not using DataInf?
>
> We believe there is no particular model configuration that limits the use of DataInf. However, as our Theorem 1 shows, the dimension of parameters can largely affect the approximation errors. Hence, parameter-efficient fine-tuning techniques (e.g., LoRA) are recommended when DataInf is applied.
>
> Thank you again for your time! We hope you would consider increasing your score if we have answered your questions. Please let us know if you have additional questions and we are happy to follow up. Thanks!

---

> > ### Author Response · Authors · 2023-11-22
> > **Looking forward to hearing from Reviewer oMqQ!**
> >
> > Dear Reviewer oMqQ,
> >
> > Thank you very much again for your time and questions on our paper. We are wondering if our rebuttal has addressed all your concerns. The discussion will end in 20 hours, but if there are any remaining or new questions, please let us know! We will be very happy to address them as immediately as we can and will improve our work.
> >
> > Best wishes,
> > Authors

---

> > > ### Comment · Reviewer_oMqQ · 2023-11-22
> > >
> > > Thanks for the response, I changed my score to 6.

---

> > > > ### Author Response · Authors · 2023-11-22
> > > > **Thank you!**
> > > >
> > > > Thank you very much for re-evaluating our work!!

---

### Official Review · Reviewer_gXtX · 2023-11-01

**Soundness:** 3 good
**Presentation:** 3 good
**Contribution:** 3 good
**Rating:** 6
**Confidence:** 3

**Summary:**

## Summary

- The paper proposes a new method to calculate Influence Scores which is more computationally and memory efficient than existing techniques like (LiSSA and Hessian Free techniques like Tracin)
- It is particularly suited for parameter-efficient fine-tuning techniques such as LoRA
- The key approximation in DataInf is to swap the order of the matrix inversion and average calculation. They conduct experiments on approximation error analysis to study the effect of this approximation because these terms are not equal in the general case. Equation (4) in the paper.
- To measure the efficacy of their proposed approximation, the authors conduct three sets of experiments: approximation error analysis, mislabeled data detection and influential data identification
	- The models used are RoBERTa model, stable-diffusion-v1.5 and LLaMA-2-13B model
	- DataInf is significantly more correlated with with exact influence values than other methods (LiSSA and Hessian free methods like Tracin) for all ranks. Correlation decreased with increasing rank which is why DataInf is specially suitable for LoRA models.
	- DataInf is better at identifying mislabeled examples
	- They also use DataInf to identify influential training examples for LLaMA-2-13B-chat model for text generation task and stable-diffusion-v1.5 model for text to image generation task.
		- They construct 3 datasets for text generation: (i) Sentence transformation (ii) Math word problems without reasoning (iii) Math word problems with reasoning
		- For text to image generation they construct two tasks (i) style generation (ii) subject generation
		- As metric, they report the percentage of training points with the same class as the test example among the top s influential training points. DataInf has significantly better recall and AUC scores than Hessian-free approach

**Strengths:**

- The result in Figure (2) is very interesting. Sometimes DataInf is even better than the exact method !

**Weaknesses:**

- Do LoRA finetuning methods typically only use low ranks like 2,4,6 as used in the experiments in the paper?

- The paper has rightly pointed out that there aren't many qualitative metrics for measuring the utility of influence scores. The authors try to address it through proxies. The results in Figure 2 are presented with only rank 4, what do the results look like for other rank values?

**Questions:**

- Curious as to what is the computational and memory complexity of Hessian free methods like Tracin which are omitted from the table?
- How did you obtain multiple training checkpoints required for techniques like tracin in your experiments? Esp for the llama model, are the checkpoints publicly released or did you do your own fine-tuning and use the fine-tuning checkpoints? Do the results for tracin improve with more checkpoints?

---

> ### Author Response · Authors · 2023-11-19
> **Response to Reviewer gXtX**
>
> Thank you very much for your feedback!
>
> > Do LoRA finetuning methods typically only use low ranks like 2,4,6 as used in the experiments in the paper?
>
> The optimal selection of low rank should highly depend on datasets and downstream tasks. However, in Table 6 of Hu et al. (2021), the original authors mentioned that a low rank (r=1,2,4, and 8) can be sufficient to achieve competitive performance for general NLP downstream tasks. Moreover, Figure 6 of our paper empirically supports this conclusion: the downstream performance does not vary much across different rank values.
>
> > The paper has rightly pointed out that there aren't many qualitative metrics for measuring the utility of influence scores. The authors try to address it through proxies. The results in Figure 2 are presented with only rank 4, what do the results look like for other rank values?
>
> Thank you for this question. The submitted manuscript actually included mislabeled data detection results for different rank values. Specifically, Figure 6 shows three different cases with the low-rank equals 1, 2, or 8. We observed a consistent pattern with different ranks: DataInf achieves better AUC than LiSSA and hessian-free methods, and sometimes better than the exact influence function. We have clarified this in the revision.
>
> > Curious as to what is the computational and memory complexity of Hessian free methods like Tracin which are omitted from the table?
>
> In terms of computational and memory complexity, the Hessian-free method has exactly the same rate as ours. We decided not to include this method as it essentially ignores the Hessian part of the influence function, yielding too crude approximation.
>
> > How did you obtain multiple training checkpoints required for techniques like tracin in your experiments? Esp for the llama model, are the checkpoints publicly released or did you do your own fine-tuning and use the fine-tuning checkpoints? Do the results for tracin improve with more checkpoints?
>
> Thank you for this question. We used only the last checkpoint of a fine-tuned model. We expect using multiple checkpoints may better describe the influence of data points, but it requires more expensive memory costs. For instance, storing all gradients of a pre-trained Llama2 model at one checkpoint requires 25 GB in our sentence transformation task. If we use $k$ checkpoints, it will require $25 \times k$ GB.
>
>
> #### Reference:
> - Hu, E. J., Shen, Y., Wallis, P., Allen-Zhu, Z., Li, Y., Wang, S., ... & Chen, W. (2021). Lora: Low-rank adaptation of large language models. arXiv preprint arXiv:2106.09685.

---

### Official Review · Reviewer_NdTy · 2023-11-02

**Soundness:** 2 fair
**Presentation:** 3 good
**Contribution:** 3 good
**Rating:** 6
**Confidence:** 3

**Summary:**

The paper proposes an efficient influence approximation method. The proposed method outperforms existing influence computation algorithms in terms of computational and memory efficiency.

**Strengths:**

- The paper tackles an important and timely problem of estimating data influence in a scalable manner.

**Weaknesses:**

- One assumption made by the paper is the first equation on page 3, which states that the expectation of hessian is equal to the expectation of the second moment of gradients. This assumption only holds true when the loss function is $-\log P(y|f_\theta(x))$, where $f_\theta(x)$ denotes the output probability of network parameterized by $\theta$. However, consider the cross entropy loss function, which is $y \log f_\theta(x)$. There is no discussion of how the assumption applies to common loss functions.

- Theorem 1 seems a loose bound. It will be much more convincing to empirically verify the introduced by the approximation and whether the error grows in $d^2$.

**Questions:**

- Section 4.1: What is the correlation analysis result when the data points are all clean?

- Section 4.2: Why does exact influence sometimes underperform the approximation methods? How does the result vary with the mislabeling ratio?

---

> ### Author Response · Authors · 2023-11-19
> **Response to Reviewer NdTy**
>
> Thank you so much for your comments.
>
> > One assumption made by the paper is the first equation on page 3, which states that the expectation of hessian is equal to the expectation of the second moment of gradients. This assumption only holds true when the loss function is $-\log P(y | f_{\theta}(x))$, where $f_{\theta}(x)$ denotes the output probability of network parameterized by $\theta$. However, consider the cross entropy loss function, which is $y \log f_{\theta}(x)$. There is no discussion of how the assumption applies to common loss functions.
>
> We would like to clarify that the cross-entropy loss function is actually a negative log-likelihood function in classification settings. To be more specific, suppose we have a $K$-class classification problem and $f_{\theta} (x) \in [0,1]^{K}$ is a probability estimate for an input data $x$. Then, the correct mathematical notation for the cross-entropy loss function is the $y$-th element of $ -\log f_{\theta} (x) $, not $ y \log f_{\theta}(x)$, when the associated label $y \in \set{1, \dots, K}$. That is, the cross-entropy loss function is a negative log-likelihood function, and for this reason, we use it in our experiments.
>
> For a general loss function, the Hessian matrix is not simplified since Bartlett’s second identity does not hold. As a result, DataInf might not be applicable to a general loss function, but we would like to emphasize that a negative log-likelihood function (i.e., the cross-entropy loss function) is the most commonly used loss function—many popular LLMs have utilized the cross-entropy loss function. We will include this discussion in the revision.

---

> ### Author Response · Authors · 2023-11-19
> **Response to Reviewer NdTy - 2**
>
> > Theorem 1 seems a loose bound. It will be much more convincing to empirically verify the introduced by the approximation and whether the error grows in $d^2$
>
> We demonstrated how the number of parameters affects the approximation error in Figure 1 of the submitted manuscript, showing that the Pearson correlation coefficient decreases when the parameter dimension increases. While it does not explicitly show the approximation error scales to $O(d^2)$, we believe it captures the key message of Theorem 1: the approximation error increases as the parameter dimension increases, and thus parameter-efficient fine-tuning techniques (e.g. LoRA) should be recommended.
>
> > Section 4.1: What is the correlation analysis result when the data points are all clean?
>
> We greatly appreciate the reviewer's constructive suggestion We have added a new correlation analysis result with a clean dataset in the revision (Figure 5). We observed that DataInf is more correlated with the exact influence function than existing methods, demonstrating the consistency of our results regardless of the presence of noisy data.
>
> > Section 4.2: Why does exact influence sometimes underperform the approximation methods? How does the result vary with the mislabeling ratio?
>
> Thank you for this question. It is expected that mislabeled data are well detected by the exact influence function as they likely have a large negative impact on validation accuracy. However, we would like to emphasize that it is not necessarily the most effective method to detect low-quality data points. The exact influence function is not guaranteed to show the best performance (as shown Jiang et al. (2023), Park et al. (2023), and Ilyas et al. (2022)), and our method shows promising results for some datasets.
>
> Thank you again for your time! We hope you would consider increasing your score if we have answered your questions. Please let us know if you have additional questions and we are happy to follow up. Thanks!
>
> #### Reference
> - Jiang, K. F., Liang, W., Zou, J., & Kwon, Y. (2023). OpenDataVal: a Unified Benchmark for Data Valuation. arXiv preprint arXiv:2306.10577.,
> - Park, S. M., Georgiev, K., Ilyas, A., Leclerc, G., & Madry, A. (2023). Trak: Attributing model behavior at scale. arXiv preprint arXiv:2303.14186.
> - Ilyas, A., Park, S. M., Engstrom, L., Leclerc, G., & Madry, A. (2022). Datamodels: Predicting predictions from training data. arXiv preprint arXiv:2202.00622.

---

> ### Author Response · Authors · 2023-11-22
> **Thank you for your review**
>
> Dear Reviewer NdTy,
>
> Thank you very much again for your time and helpful feedback on our paper. In particular, we are pleased that our new experiment, which has been suggested by the reviewer, finds that DataInf is highly correlated to the exact influence function even when a dataset is clean (Figure 5).
>
> We are wondering if our rebuttal has addressed all your concerns. The discussion will end in 20 hours, but if there are any remaining or new questions, please let us know! We will be very happy to address them as immediately as we can and will improve our work.
>
> Best wishes,
> Authors

---

### Author Response · Authors · 2023-11-19
**Response to all reviewers**

We thank all the reviewers for their time and helpful comments. In the revised paper that we have uploaded, we have carefully incorporated the reviewers’ suggestions, and changes have been colored in red. Also, we respond to each of the reviewer’s comments in separate posts.

---

### Meta-Review · Area_Chair_rQaQ · 2023-12-07

**Metareview:**

This paper proposes an efficient approximation of the influence function that can be used on generative AI models.

**STRENGTHS**

(1) The proposed approximation method is more efficient than existing ones.

(2) This paper addresses an important and timely problem.

(3) The reviewers find the work and results interesting.


**WEAKNESSES**

The authors have done a good job addressing the reviewers' concerns. Reviewer NdTy has also responded positively with an increase in the rating.

However, there are still some remaining issues that can be readily addressed:

(1) As mentioned by Reviewer NdTy, Fig. 1 does not explicitly show how the approximation error changes directly with the number $\sum_l d_l$ of neural network parameters (but rather the rank r of the LoRA matrix). Such important empirical results are missing.

(2) From the rebuttal, it indeed remains unclear why the exact influence underperforms the approximation methods. A deeper detailed analysis and discussion would be necessary here to understand this.

(3) The related work section is missing quite a fair bit of existing literature on non-influence-based data valuation methods, especially the efficient training-free ones. See the following references/links and those therein:

Data Valuation Without Training of a Model. ICLR 2023.

DAVINZ: Data Valuation using Deep Neural Networks at Initialization. ICML 2022.

Data Valuation in Machine Learning: "Ingredients", Strategies, and Open Challenges. IJCAI 2022.

https://github.com/daviddao/awesome-data-valuation


The authors are strongly encouraged to address the above remaining issues and include their clarifications in the rebuttal in their revised paper.

**Justification For Why Not Higher Score:**

There are remaining issues that can be readily addressed by the authors. There is also no strong advocate for a higher rating for this paper.

**Justification For Why Not Lower Score:**

All the reviewers have agreed with the nontrivial contributions of this work, as highlighted in the strengths.

---

### Decision · Program_Chairs · 2024-01-16

Accept (poster)